# Robust Amyloidosis Subtype Classification via Multisequence CMR Fusion with Spatiotemporal Learning

**Parker Martin**[1]                                     Parker.Martin@osumc.edu
**Yifan Liu**[1]                                         liu.11275@buckeyemail.osu.edu
**Karolina M. Zareba**[1]                                Karolina.Zareba@osumc.edu
**Suzanne Smart**[1]                                     Suzanne.Smart@osumc.edu
**Akash Goyal**[1]                                       Akash.Goyal@osumc.edu
**Orlando P. Simonetti**[1]                              Orlando.Simonetti@osumc.edu
**Jeremy Slivnick**[2]                                   jslivnick@bsd.uchicago.edu
**Yuan Xue**[1]                                          Yuan.Xue@osumc.edu

[1] *The Ohio State University, Columbus, OH, USA*
[2] *University of Chicago, Chicago, IL, USA*

**Editors:** Accepted for publication at MIDL 2025

## Abstract

Cardiac amyloidosis (CA) subtype classification is a critical diagnostic challenge. We propose a multimodal deep learning framework that integrates cine, late gadolinium enhancement (LGE), and T1/T2 parametric cardiac MRI sequences to differentiate AL and ATTR amyloidosis. Its sequence-specific encoders and gated attention fusion enable robust performance, even with missing inputs. Evaluated on 123 patients with cross-validation, the xLSTM-based model achieved the highest AUC (0.8506), outperforming a Video Swin Transformer (VST). Grad-CAMs highlight cardiac and extracardiac regions, showing interpretability and potential for identifying systemic imaging biomarkers. Results support a clinically viable non-invasive CA subtype diagnostic approach.

**Keywords:** Cardiac Amyloidosis, Cardiac MRI, Deep Learning, Multimodal Learning, Spatiotemporal Modeling, Model Interpretability

## 1. Introduction

Cardiac amyloidosis (CA) is a progressive infiltrative cardiomyopathy caused by extracellular deposition of misfolded proteins, commonly light chain (AL) or transthyretin (ATTR). Accurate subtype classification is key for therapy and prognosis (Maggialetti et al., 2024; Aus dem Siepen and Hansen, 2024), but current pathways often use invasive biopsy or specialized nuclear imaging (Dorbala et al., 2020). Cardiac MRI (CMR) offers a non-invasive, information rich alternative (Fontana et al., 2015a), yet interpreting cine, LGE, and T1/T2 maps requires expertise and remains challenging (Fontana et al., 2015b; Banypersad et al., 2015; Zhao et al., 2016). We present a multimodal deep learning model combining these CMR sequences via sequence-specific encoders and gated attention fusion (Azam et al., 2022), with a focus on spatiotemporal learning from cine using xLSTM (Beck et al., 2024). The model handles missing sequences and offers interpretable Grad-CAMs (Selvaraju et al., 2017), for robust, explainable CA subtype classification.

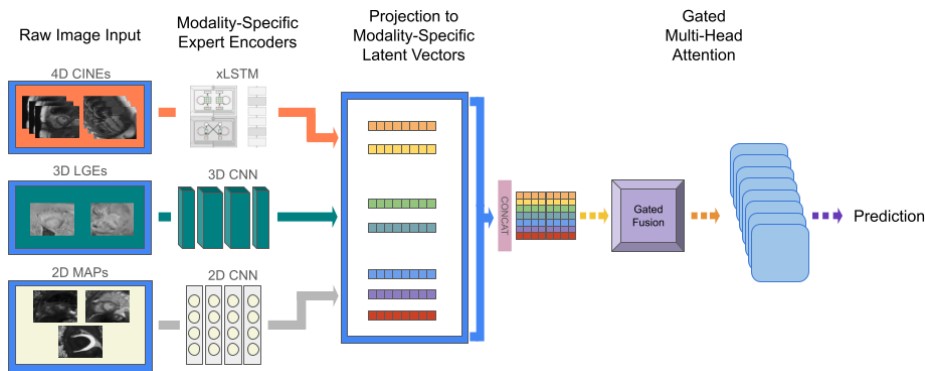

Figure 1: Model: sequence-specific encoders & gated multi-head attention fusion.

## 2. Methodology

**Dataset and Preprocessing.** Cardiac MRI scans from 123 biopsy-confirmed amyloidosis patients at The Ohio State University were used. Each exam included cine sequences (2D time-resolved), LGE (multi-slice 2D stacks), and T1/T2 parametric maps. Acquired as 2D/3D clinically, dimensionality refers to model input structure. All images were resampled to 0.94 mm isotropic resolution. Intensity normalization used z-scoring for cine and LGE, and percentile scaling for parametric maps.

**Model Architecture.** Our framework (Fig. 1) employs sequence-specific encoders: a spatiotemporal network via xLSTM (Beck et al., 2024) or Video Swin Transformer (Liu et al., 2022) for cine MRI, a 3D CNN for LGE, and a 2D CNN for parametric maps. Each encoder outputs a latent representation, which is dynamically fused using gated multi-head attention (Vaswani et al., 2017; Bahdanau et al., 2014). This design supports robustness to missing modalities while preserving temporal structure in cine sequences.

**Training Strategy.** We conducted 5-fold stratified cross-validation to compare model variants: (1) xLSTM vs. VST for cine encoding, and (2) with or without demographics (age, sex). Single-sequence baselines were also evaluated. Models were trained with cross-entropy loss, Adam optimizer (Kingma and Ba, 2015), dropout, and label smoothing. Hyperparameters were selected via Bayesian optimization using TPE (Bergstra et al., 2013).

## 3. Results

**Model Comparison.** Cross-validation results (Table 1) show that the spatiotemporal xLSTM achieved the highest AUC ($0.8506 \pm 0.0654$) with lower computational cost. VST offered slightly lower accuracy ($0.8346 \pm 0.0410$) at nearly double the parameter count and memory usage. Including demographic data modestly reduced performance in both models

Table 1: Cross-validation performance comparison across configurations.

| Model Configuration | AUC | Inference Time (ms) | Min/Med/Max (ms) | # Params | Peak GPU (MB) |
|---|---|---|---|---|---|
| Spatiotemporal xLSTM | $0.8506 \pm 0.0654$ | $65.42 \pm 45.50$ | 46.12 / 53.73 / 283.31 | 137M | 71.67 |
| Video Swin Transformer | $0.8346 \pm 0.0410$ | $107.31 \pm 36.63$ | 89.88 / 97.31 / 309.29 | 276M | 97.76 |
| xLSTM + Demographics | $0.8244 \pm 0.0820$ | $61.86 \pm 44.57$ | 45.06 / 50.26 / 275.59 | 137M | 71.67 |
| VST + Demographics | $0.8154 \pm 0.0656$ | $110.79 \pm 37.10$ | 93.67 / 101.56 / 298.95 | 276M | 97.76 |

Figure 2: Grad-CAMs: myocardial/extracardiac (e.g., kidneys) activations, suggesting systemic features for subtype differentiation.

(AUCs 0.8244 and 0.8154). This reduction might stem from the limited dataset size or challenges in encoding demographics relative to rich image features.

**Single-Sequence Instability.** Models trained on individual sequences failed to converge reliably across folds (precluding quantitative comparison), regardless of architecture. This instability reinforces the value of multimodal fusion, suggesting that complementary information across sequences is essential for subtype discrimination.

**Interpretability and Systemic Insights.** Fig. 2 displays Grad-CAM visualizations showing myocardial and extracardiac activations. Attention weights were consistently balanced across modalities (range: 0.60–0.65). Notably, extracardiac signals frequently appeared in renal regions, consistent with systemic amyloid burden reported in radiological literature (Kawashima et al., 2011).

## 4. Discussion and Conclusion

Our study shows a multimodal deep learning framework (cine, LGE, mapping) achieves strong CA subtype classification. The xLSTM encoder offers a good accuracy/cost balance, suitable for clinical deployment.

Demographics did not aid classification, possibly due to limited sample size, challenges integrating them with rich image features, or dominant imaging features. Moreover, extracardiac Grad-CAMs may reflect systemic amyloid burden, opening research into holistic biomarkers.

These findings align with prior work on CMR tissue characterization (Fontana et al., 2015a; Banypersad et al., 2015). We extend this by providing an interpretable framework learning from incomplete/heterogeneous inputs.

In conclusion, our spatiotemporal model shows strong, interpretable CA subtype classification with multisequence MRI. Future work will include multi-institutional external validation, sophisticated fusion of demographic and lab data, deeper systemic feature investigation, and comparisons with established clinical diagnostic algorithms.

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
