# OpenReview forum: "Robust Amyloidosis Subtype Classification via Multisequence CMR Fusion with Spatiotemporal Learning"
_MIDL.io/2025/Short_Papers — MIDL 2025 - Short Papers_

### Official Review · Reviewer_GTJJ · 2025-04-28

**Rating:** 4
**Confidence:** 5

**Summary:**

This study focuses on the automatic classification of cardiac amyloidosis (CA) subtypes.
The originality of the method lies in the integration of different MR sequences (cine, late gadolinium enhancement, T1, and T2) to differentiate between two types of amyloidosis. The proposed solution was evaluated on 123 patients using cross-validation.
Besides the classical performance evaluation based on AUC, the authors also investigated the interpretability of the results through an analysis of attention on the images using the Grad-CAM method.

**Strengths:**

The strengths of this work are:
1) the relevance of the topic of this study, any improvement/innovation of which could have a major impact in our field
2) the relevance of the experiments.
3) the richeness of the dataset (cine, LGE, T1/T2 for each patient)
4) the quality of the results

**Weaknesses:**

The main weaknesses of this article concern:
1) It is mentioned that "models trained on individual sequences failed to converge reliably across folds, regardless of architecture"; however, no results for this category of methods were provided.
2) It is observed that including demographic data modestly reduced performance in both models, with the explanation that this was likely due to overfitting given the limited dataset size. However, this explanation appears overly simplistic. One possible reason could be that the demographic information was poorly encoded relative to the richness of the image latent space, which is a known issue.
3) comparisons with other methods are needed to draw more relevant conclusions

---

### Decision · Program_Chairs · 2025-05-01

Accept